# Paradoxes in leaky microbial trade

Yoav Kallus [iD] [1], John H. Miller[1,2] & Eric Libby[1]

Microbes produce metabolic resources that are important for cell growth yet leak into the environment. Other microbes can use these resources, adjust their own metabolic production accordingly, and alter the resources available for others. We analyze a model in which metabolite concentrations, production regulation, and population frequencies coevolve in the simple case of two cell types producing two metabolites. We identify three paradoxes where changes that should intuitively benefit a cell type actually harm it. For example, a cell type can become more efficient at producing a metabolite and its relative frequency can decrease—or alternatively the total population growth rate can decrease. Another paradox occurs when a cell type manipulates its counterpart's production so as to maximize its own instantaneous growth rate, only to achieve a lower final growth rate than had it not manipulated. These paradoxes highlight the complex and counterintuitive dynamics that emerge in simple microbial economies.

[1] Santa Fe Institute, 1399 Hyde Park Road, Santa Fe, NM 87501, USA. [2] Carnegie Mellon University, 5000 Forbes Avenue, Pittsburgh, PA 15213, USA. Correspondence and requests for materials should be addressed to E.L. (email: elibby@santafe.edu)

Microbes live in complex communities where goods such as metabolites are produced and exchanged[1–4]. As goods flow in and out of cells, a type of economy emerges[5, 6]. In this economy, each organism faces decisions concerning which goods to produce and in what quantities[7]. These production decisions ultimately determine the relative abundance of each organism since more successful individuals will grow faster and increase in frequency. As populations change, the economic conditions can change and put pressure on organisms to adjust their production[6]. In this paper, we investigate this interplay between population-level dynamics and individual-level production decisions and uncover paradoxical system-level behaviors.

Microbes exchange goods directly or indirectly[8]. Direct mechanisms, such as intercellular nanotubes[9] or cell–cell recognition systems[3], allow microbes to target goods towards specific partners, thereby facilitating successful trading relationships. In contrast, indirect exchange typically relies on the diffusion of molecules through the extracellular environment[8, 10]. Some goods are produced and secreted because their primary function occurs extracellularly. One classic example is a siderophore that binds extracellular iron and allows it to be imported into the cell[11]. Other goods diffuse out of cells through inherently permeable cell membranes[8]. Metabolic by-products and electron carriers are examples of these kinds of leaked goods[8, 12]. Once such goods are in the environment, they can be used to inform individual production strategies. Here, we focus exclusively on indirect exchange of goods via diffusion.

Even if we consider only indirect exchange of diffusible goods, there is a great diversity of the types of exchange depending on the environmental and ecological context, the number of organisms and goods, as well as the costs and benefits of the goods[10, 13]. We narrow our scope by considering only interactions between two organisms involving two goods. This excludes well-studied systems of trade such as the mutualism between mycorrhizal fungi and plants in which many different organisms may be trading simultaneously[14, 15]. Furthermore, we only consider goods that are costly to make and are beneficial to at least one organism. Thus, we do not consider punitive goods such as toxins or antibiotics. We find it useful to classify goods in terms of which organisms produce them and which organisms benefit from their consumption. Using this approach, Table 1 shows three canonical types of exchange. For each good, we denote which organism produces it ($p$) and which organism consumes it ($c$). The three types of exchange do not represent an exhaustive classification, but rather provide a way of comparing exchange interactions that have received significant attention in previous studies.

The first category is mutualism, where each organism produces a good that the other one consumes. This type of relationship can represent syntrophy[13, 16], cross-feeding[17], auxotrophy[4], or a two-way by-product mutualism[18, 19]. Since each organism does not consume the good that they produce, the goods are by-products of other processes. This means that the optimal amount of the by-product to produce depends on the costs and benefits of the other, more primary processes, as well as how

much benefit is derived from the good produced by the other organism[17, 19, 20]. In instances where each good produced is growth-limiting to the other organism, there is a positive feedback loop so that each organism does its best by producing as much of their good as possible, so long as it does not interfere with other cell functions. One common result of these syntrophic exchanges is a synergy between organisms, where the combined community has enhanced growth relative to any isolated individual[6].

In the second category, exploitation, one organism produces a good that both organisms value, while the other organism produces only goods of value to itself. This arrangement captures parasitic behaviors as well as forms of cheating and competition[17, 21]. Indeed, this arrangement describes the public goods dilemma that has been well-studied in social evolution[11]. Although one organism is exploiting the other, there is no real production decision for the producer since it needs the good and is the only one that produces it. This situation is at the heart of the Black Queen Hypothesis, where adaptive gene loss leaves one organism burdened with producing a costly metabolite that is exploited by the community[22].

The final category, self-sufficiency, represents the most flexible and possibly primitive arrangement. Here, each organism is capable of producing all of the goods it needs for survival and both organisms value these goods. The possible goods that fit this scenario include amino acids or molecules that are essential to central metabolism or maintenance. Interestingly, this category is a precursor to the other categories, as loss-of-function mutations can result in either mutualism or exploitation scenarios[23]. Thus, the self-sufficiency case is often the starting point for models that explore the Black Queen Hypothesis[10, 23]. In this paper, we focus exclusively on the self-sufficiency arrangement in order to understand its dynamics and how they might prime populations to evolve into one of the other categories.

The self-sufficiency case has also been studied implicitly in models of metabolic trade. In these models, metabolic networks that are capable of growing on a variety of resources, are joined together to understand how the combined metabolism might function[24–26]. The production decisions are solved using some objective function and flux balance analysis. By joining metabolisms, it has been shown that extant organisms can grow on a wide variety of resources[27]. One feature lacking in these models is the dynamic interplay between population composition and production—especially when organisms have different production capabilities and there is a tension between maximizing individual and population growth rates.

Here, we address the issue of population composition and growth with a general microbial trade model that couples population dynamics to organism production strategies. We assume that each organism alters its production in order to maximize its own growth rate. Since microbes can shift the production of costly goods depending on environmental concentrations, each organism's production of leaky goods affects the production strategies of other organisms. Using this approach, we uncover three unusual system-level behaviors that apply to the

**Table 1 Classification of microbial exchange between two organisms involving two goods**

|        | Mutualism | | Exploitation | | Self-sufficiency | |
|--------|------------|------------|------------|------------|------------|------------|
|        | Organism 1 | Organism 2 | Organism 1 | Organism 2 | Organism 1 | Organism 2 |
| Good 1 | $p$ | $c$ | $p,c$ | $c$ | $p,c$ | $p,c$ |
| Good 2 | $c$ | $p$ | — | $p,c$ | $p,c$ | $p,c$ |

A $p$ indicates that a good is produced and a $c$ indicates that it is consumed. We assume that if an organism consumes a good, it benefits in some way

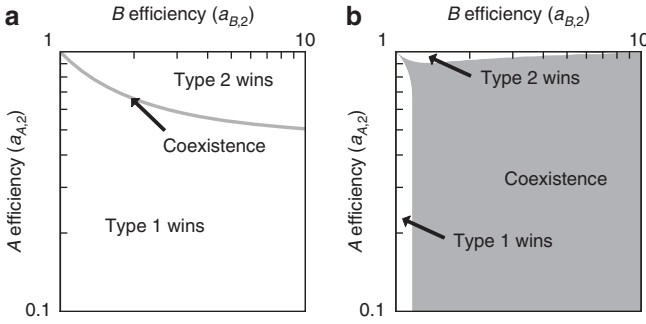

**Fig. 1** Coexistence with and without diffusion. **a** The growth rate of a population consisting of a single cell type is a function of the maximum amount of each essential metabolite that it can produce, i.e., $a_A$ for A and $a_B$ for B. The gray line is the locus where the growth rate of a cell (type 2) equals that of a reference cell (type 1) that has equal costs for producing either metabolite, with $a_{A,1} = a_{B,1} = 1$. In the absence of diffusion, coexistence with the reference cell type is only possible where the growth rates are equal. Above the gray line, the type 2 cell grows faster and drives the type 1 cells to zero relative frequency, i.e., the type 2 cells "win." Below the line, the situation is reversed. The coexistence line represents a transition between the different cell types winning. **b** With diffusion, e.g., when $D = 3$, each cell-type population is affected by the other's production. The coexistence region is significantly larger and fills much of the quadrant considered, corresponding to where each cell type is more efficient than the other at producing one of the metabolites

relevant trading scenarios between microorganisms. Furthermore, these behaviors suggest evolutionary trajectories that lead populations to more structured forms of arrangement such as mutualisms or exploitations.

## Results

**Extinction and coexistence.** We consider a microbial population model in which organisms trade through the production and diffusion of metabolites. For simplicity, we assume that there are two types of organisms (1 and 2) that require the same two metabolites (A and B) in order to grow and reproduce. We denote the amount of A and B metabolites in cells of type $i = 1$, 2 as $A_i$ and $B_i$ and the proportion of cells of type $i$ by $n_i$. Since we choose to analyze the self-sufficiency case in Table 1, each organism can consume and produce both A and B metabolites. Production, however, comes with costs either as a result of energy expenditure or forfeited opportunities to produce other goods or engage in other processes. We assume that the production rates of metabolites are subject to a budget constraint, whereby the organism has a finite amount of resources (precursors, enzymes, ribosomes, etc.) that can be devoted to the production of metabolites (Methods section, Eq. (1)). Besides consumption and production, metabolites can be gained or lost through passive diffusion depending on the concentration gradient across the cell membrane. The diffusive flux is mediated via the coefficient $D$. Finally, metabolites are lost at a rate $\mu$ either due to diffusion away from the shared environment or by some process of degradation.

Before we explore the behavior of interacting microbial populations, we first consider the growth of a population of cell types in the absence of trade. We prevent different cell types from exchanging metabolites by setting $D = 0$ in Eq. (2). We assume that every cell regulates its production rates of metabolites $p_{A,i}$ and $p_{B,i}$ so as to maximize its own growth. Because our production constraint function Eq. (1) does not feature returns to scale, there is no benefit to a division of labor among members of the same cell type. As a consequence, every cell of the same

type shares an identical strategy in terms of how much of each metabolite is produced. We compute the growth rate for a cell type as a function of the energetic costs of making A and B metabolites, or equivalently, the inverse of the costs. We call the inverse of a production cost the efficiency, that is, $a_{X,i} = 1/c_{X,i}$, and it corresponds to the maximum amount of the good a cell can produce.

In Fig. 1a, the gray line shows production efficiencies that yield the same growth rate as a reference cell type, say cell type 1, that is equally efficient at producing either metabolite, with $a_{A,1} = a_{B,1} = 1$. When we add cell type 2 with production efficiencies given by $a_{A,2}$ and $a_{B,2}$ to a population of the reference cell type, the only way for the two cell types to coexist is if the production efficiencies lie on the gray line; otherwise, one of the two cell types will grow faster and tend to 100% of the population. Type 2 cells with efficiencies above the gray line in Fig. 1a grow faster than the reference cell type and will ultimately drive it extinct; below the gray line, the reverse is true. Thus, in the absence of diffusion, coexistence is rare and only possible for a specific relationship between production efficiencies that constrains the two cells types to grow at equal rates.

We now consider what happens when two populations of cell types can exchange metabolites (i.e., $D > 0$). For any initial mixed population, $n_1$, $n_2 > 0$, there is a Nash equilibrium choice of production rates where neither organism can improve its growth rate by changing its production. The growth rates of each cell type at this Nash equilibrium are not necessarily the same. If the growth rates are different, then one cell type will increase in relative frequency. This will alter the relative frequencies of the two cell types $n_1, n_2$ and could lead cell types to adapt to the new frequencies by changing their production. This process continues until either the growth rates of the two cell types are equal and their relative frequencies are stable or one cell type is driven towards extinction (zero frequency). For our choice of growth functions, production constraints, and parameters, there is always a unique stable equilibrium $n_1^*$ in terms of the relative frequencies of cell types. This means that for a given set of metabolic efficiencies, all mixed populations will approach the same equilibrium of relative frequencies. Of course, a change in the efficiency of producing a metabolite could alter this equilibrium.

We compute the equilibrium $n_1^*$ as a function of the relative efficiencies of producing metabolites A and B. As before, we hold one cell type, $i = 1$, fixed in terms of its efficiencies and vary the efficiencies for the other cell type, $i = 2$. When one cell type is better than the other at both production tasks, the only stable equilibrium is that its fraction of the population approaches one, and the other cell type is driven to extinction (results not shown). This trivial result seemingly contradicts the notion of a comparative advantage, familiar from Ricardian economics[28], where there is a benefit from trade even if one agent is better at producing all goods. In fact, at a fixed value of the relative frequency $0 < n_1 < 1$, a comparative advantage does play a role in setting the Nash equilibrium, and both cell-type populations benefit from the diffusive exchange. However, because cells can reproduce, if one cell type, say 1, is better than the other at both production tasks, it will always grow faster, and any relative frequency except $n_1 = 1$ will not be sustainable. For the rest of the paper, we ignore this case and consider instead the case where each cell type is more efficient than the other at producing one of the two metabolites.

In Fig. 1b, the gray region indicates where a coexistence equilibrium is observed. This region is much expanded in comparison to the line in Fig. 1a, and many more combinations of efficiencies lead to coexistence. Even though neither cell type is more efficient than the other in the production of both metabolites, there is still a region of parameter space in which

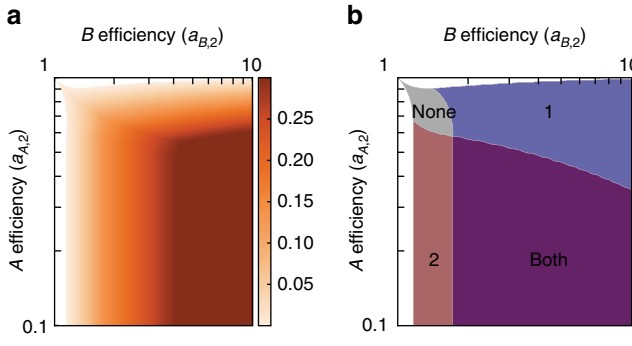

Fig. 2 Specialization and the benefit of trade. **a** The equilibrium growth rate of a population of two coexisting cell types is larger than what either of them would be able to achieve alone. Here, we plot the difference between the growth rate of a population of two cell types (one with the efficiencies shown and a reference cell type with $a_{A,1} = a_{B,1}$) and that of the surviving cell type in the absence of diffusion. All areas of coexistence in the two-cell-type population grow faster than the clonal cell population. **b** The higher growth rate is achieved by cell types shifting the production toward the metabolite in which they have a higher efficiency than the other cell type. This shift may be complete for both types (purple), only for cell type 1 (blue) or 2 (red), or for neither (gray)

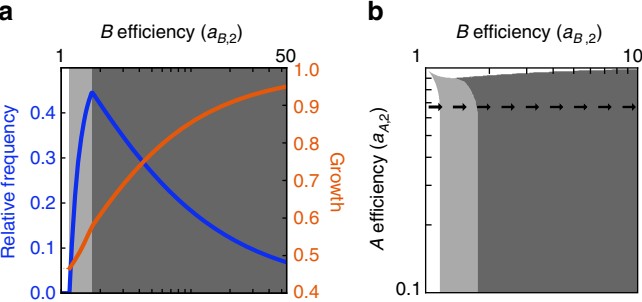

Fig. 3 The curse of increased efficiency. **a** The relative frequency of type 2 cells (blue) and growth rate of the total population (orange) are shown as a function of the efficiency of type 2 cells in producing metabolite B. We fix $a_{A,2} = 0.67$. As the efficiency of type 2 cells increases, their relative frequency ultimately decreases. The population growth rate, however, increases with a higher metabolic efficiency. **b** The shaded regions indicate where type 2 cell populations are increasing (light gray) or decreasing (dark gray) in relative frequency as they improve in efficiency in producing metabolite B, that is, moving from left to right along the indicated line. In the dark-gray region, the relative frequency of cell type 2 decreases toward 0 as its efficiency increases toward infinity

there is failure of coexistence. For example, when cells of type 2 are significantly worse at producing $A$ than their counterparts but are only marginally better at producing $B$, then, the system tends toward an equilibrium where cell type 2 goes extinct ($n_2 \to 0$). Similarly, there is a corresponding region where cells of type 2 are significantly better at producing $B$ but are only marginally worse at producing $A$, and they take over the population ($n_2 \to 1$).

Where coexistence occurs, we find that the growth rate (equal for the two cell types, by definition of the equilibrium) is larger than either cell type would have been able to achieve in isolation (Fig. 2a). By concentrating the production to the metabolite each cell type is better at producing, both cell types experience an increased growth rate. This result has been found in other, different models of microbial trade[6, 29]. In our model, the advantage of trade is achieved even when specialization is not complete, that is, when a cell type produces both metabolites. In Fig. 2b, we show the regions in parameter space where either both, one, or neither of the cell types specialize completely. In general, the highest growth rates occur where both cell types completely specialize, though there are regions of high growth where only one cell type completely specializes. In all cases, the increased growth rate resulting from trade and specialization, that is, compared to growth in isolation, does not require any global coordination between the cell types. Rather, it emerges from each cell type producing what maximizes its own growth rate.

Until now, we have primarily investigated what conditions permit coexistence. However, if we also consider the resulting population composition and growth rates, we find that the interplay between the three types of dynamic variables in our model can lead to seemingly paradoxical phenomena. We illustrate three salient examples below.

**Paradox 1: the curse of increased efficiency.** The first paradox concerns the relative frequency of cells of type 2 as a function of their metabolite production efficiency. Specifically, we consider a horizontal cross section of the parameter space in Fig. 2b where $a_{A,2}$ is fixed and $a_{B,2}$ varies. As cells of type 2 become better and better at producing $B$, their relative frequency at first increases as might be expected due to their increased productivity. However,

at some point, their relative frequency reaches a maximum and declines (Fig. 3a). Thus, even though the type 2 cells can produce more of the $B$ metabolite without decreasing the production of $A$, they represent a smaller fraction of the population. This effect intensifies as the production efficiency of $B$ increases toward infinity, driving the fraction of type 2 cells in the population toward 0.

The region where this "curse" is in effect is illustrated in Fig. 3b. It occurs in the area where cell type 1 is completely specialized and only makes the $A$ metabolite. Since cells need both $A$ and $B$ to grow, cell type 1 specializes in $A$ because there is enough $B$ in the environment provided by cell type 2 for it to forego production of $B$. Along a horizontal cross section, following the arrows in Fig. 3b, cell type 2 gets more efficient at producing $B$, but cell type 1 has the same production capacity for $A$. Because type 1 cells are not getting better at producing $A$, their fraction of the population—needed to support the continued growth of both populations—increases. This phenomenon can be seen analytically in the limit of a small degradation rate $\mu$ and under the assumption of full specialization, where $p_{B,1} = p_{A,2} = 0$. In this case, we have $n_1 p_{A,1} = n_2 p_{B,2}$ and so, $n_2 = a_{A,1}/(a_{A,1} + a_{B,2})$, where the fraction of type 2 cells is inversely proportionate to their efficiency in producing $B$.

Though the relative frequency of type 2 cells increases and decreases as $a_{B,2}$ increases, the population growth rate is always increasing. Therefore, even when the relative frequency of cell type 2 is decreasing, its total numbers might not be decreasing, because it is growing at a faster rate than it would be otherwise. The local effects—decreased relative frequency of cell type 2—are paradoxical, but the global effects—the population growth rate—are consistent with expectations. This observation partially resolves the paradox wherein a cell-type population appears to suffer as a result of a gain in efficiency. However, in the next paradox, we will show that even when considering the growth rate instead of the relative frequency, a population can suffer as a result of a gain in efficiency.

**Paradox 2: the curse of decreased inefficiency.** The second paradox concerns the population growth rate of both cell types as

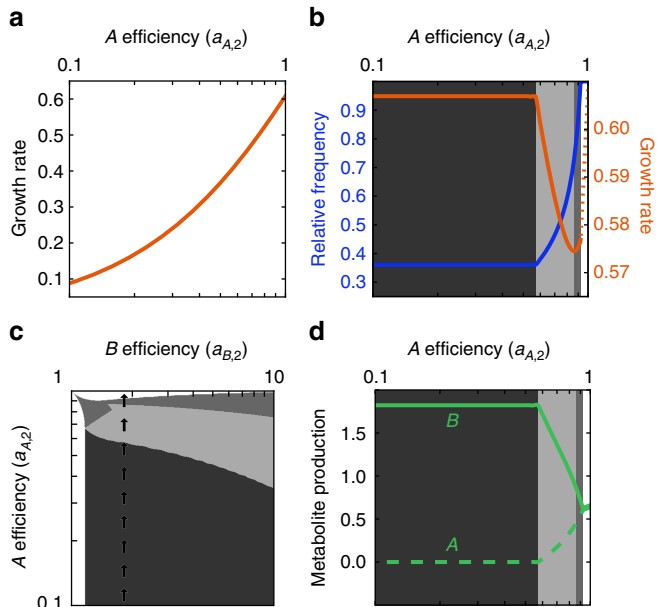

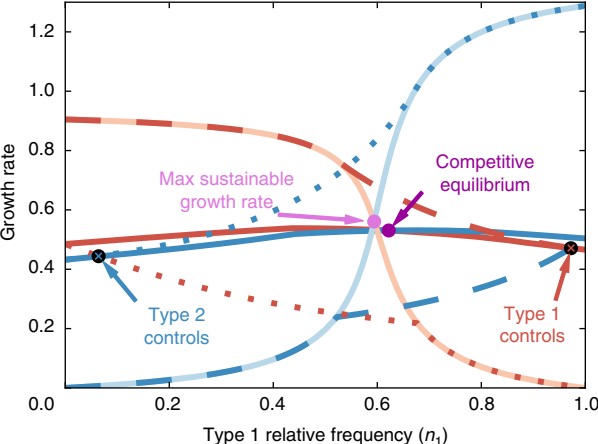

**Fig. 4** The curse of decreased inefficiency. **a** The growth rate of type 2 cells in isolation increases as they improve in efficiency of making $A$, the metabolite that they produce more poorly ($a_{B,2} = 1.82$, $a_{A,2} < 1$). **b** Similar to the previous plot, but in the presence of a reference cell type ($a_{B,1} = a_{A,1} = 1$). Here, as the type 2 cells increase in efficiency of producing $A$, the population growth rate (orange) decreases in the light-gray region. This also corresponds to an increase in relative frequency of the type 2 cell (blue). **c** The shaded regions indicate where the population growth rate is constant (dark gray), decreasing (light gray), or increasing (middle gray) as type 2 cells improve in efficiency in producing metabolite $A$, i.e., moving up along the indicated line. **d** The amount of $B$ and $A$ metabolites produced by type 2 cells is shown as a function of the efficiency in producing $A$. In the light-gray region, where the population growth rate decreases, the type 2 cells shift the production from the $B$ metabolite to the $A$ metabolite

**Fig. 5** Different types of equilibria and the growth cost of control. The steady-state growth rate of cells of type 1 (red) and type 2 (blue) under four different scenarios: each type maximizing its own growth (dark solid lines), both types maximizing the growth of cells of type 1 (dashed), both types maximizing the growth of cells of type 2 (dotted), and perfect coordination, where each cell type produces only a single metabolite and the population achieves the maximum sustainable growth rate (light solid lines). The production efficiencies used in this example are $a_{A,1} = a_{B,1} = 1$, $a_{A,2} = 0.67$, and $a_{B,2} = 1.49$. The resulting population dynamic equilibria are marked. The growth rate of the competitive equilibrium is closer to the maximum sustainable growth rate than either equilibria reached when a single cell type is in control

a function of metabolite production efficiency. Here, we consider a vertical cross section of Fig. 2b, where type 2 cells have a fixed efficiency of producing $B$, but a varying efficiency of producing $A$. Traversing up a vertical cross section corresponds to type 2 cells being able to produce more $A$ but still not as much as type 1 cells. In a homogeneous population with only one cell type, any improved efficiency in production would correspond to an increased growth rate (Fig. 4a). However, in a mixed population, Fig. 4b shows that as cells of type 2 get more efficient at producing the $A$ metabolite, the population growth rate decreases (before ultimately increasing). Thus, despite an increased capacity to produce metabolites, the population grows more slowly.

To explain this paradox, we consider the absolute maximum growth rate that a population of two coexisting cell types could achieve, assuming that they perfectly coordinated their production of metabolites. For this maximum growth rate to be sustainable, the growth rates of both cell types must be equal; otherwise, the relative frequency of the cell types will change and the population will no longer be able to sustain this growth rate. There is no reason for the maximum sustainable growth rate to be achieved as a Nash equilibrium, and in general it is not. We determine the parameters $n_1$, $p_{A,1}$, $p_{B,1}$, $p_{A,2}$, and $p_{B,2}$ that correspond to the maximum sustainable growth rate and find that in almost all cases, this optimum is achieved when cell types fully specialize in their production of metabolites, that is, $p_{B,1} = p_{A,2} = 0$ and $p_{B,2}$, $p_{A,1} > 0$. The reason, then, that increasing the efficiency of cell type 2 to produce $A$ decreases the growth rate

of the population is that it moves the Nash equilibrium away from complete specialization, that is, away from the steady state that achieves the maximum sustainable growth rate. This explains why the population growth rate starts decreasing at the same point that cell type 2 no longer specializes (Fig. 4c, d). Interestingly there is a small range for $a_{B,2}$ for which there are two cycles of decreasing and increasing the growth rate, corresponding to type 2 cells shifting the production and then type 1 cells shifting the production (Supplementary Fig. 3).

**Paradox 3: the curse of control**. The final paradox focuses on the population growth rate as a function of how metabolic production is determined. Until now, we have assumed that both cell types are choosing their own production, so as to maximize their own growth rate. Here, we consider what happens if one cell type is able to determine both its own production rates and those of the other cell types. This situation occurs in some game-theoretic settings[30], where a single player can force others to follow a particular strategy of their choice, for example, by playing a punitive strategy when the other players deviate. In the case of microbes, we imagine that a microbial population has evolved the ability to manipulate its partner. While this scenario is mostly a theoretical construct, it is inspired by the ability of microbes to manipulate quorum-sensing systems[31] as well as the ability of parasites to manipulate their host's behavior[32].

We implement the manipulation by assuming that the production rates for both cell types are chosen to maximize the growth rate of type 1 cells, regardless of the resulting growth rate for cells of type 2. We repeat the numerical process as before where we solve for the steady-state growth rates at a given value of $n_1$, and depending on the relative value of growth rates, they either increase or decrease $n_1$. Since cell type 1 is controlling the production, it continually increases in relative frequency until the growth rates of the two cell types are identical. We find that while the resulting equilibrium has a larger proportion of cells of type 1

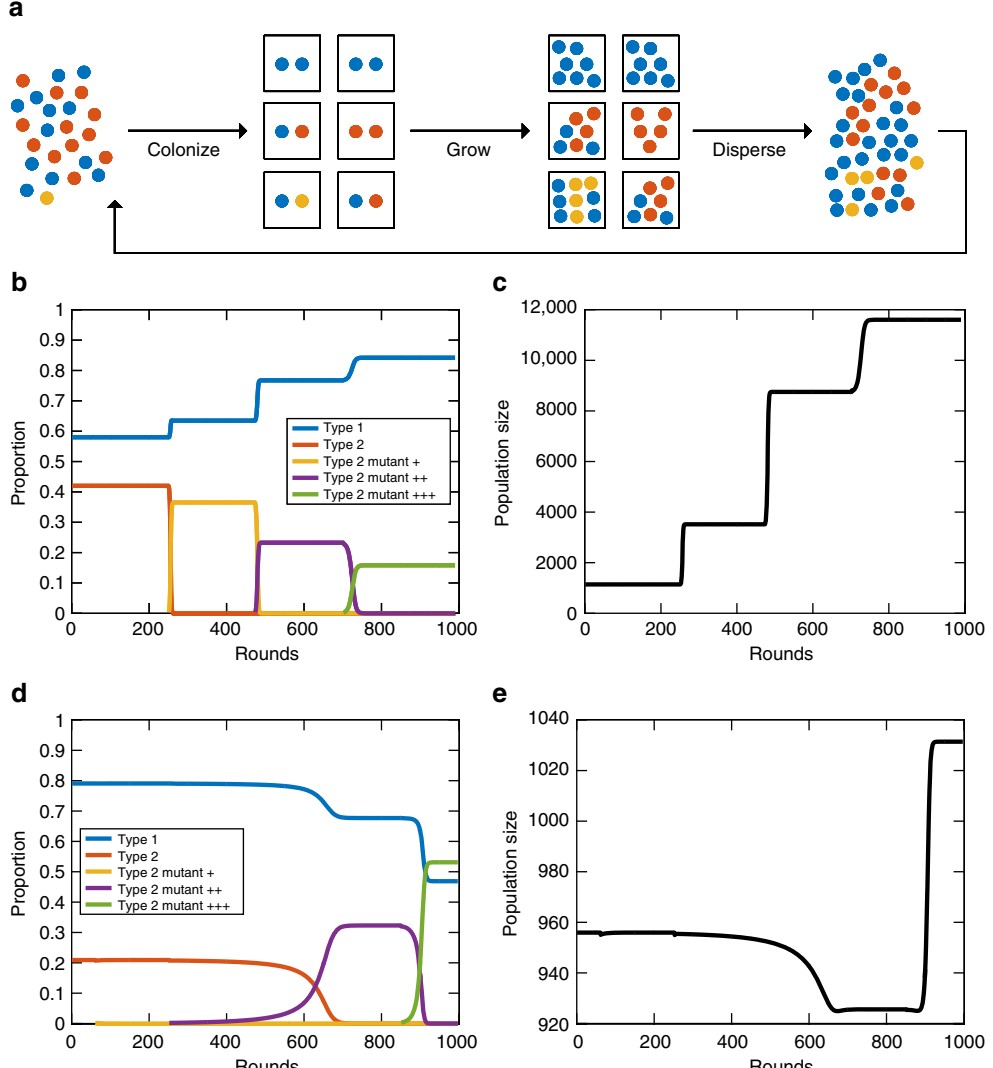

**Fig. 6** Evolution of paradoxical behavior in a metapopulation model. **a** A schematic shows the structure of the metapopulation model. Each patch (box) is colonized by two cells that are potentially of the same type. The cells grow and reproduce in the patch for a fixed amount of time and then, they are released to colonize new patches, thereby completing a "round". The process is repeated until a steady state of relative frequencies is reached. **b** Paradox 1 is observed when cell type 2 mutants with increased $a_{B,2}$ invade. Each + indicates a mutant with an increased efficiency over its ancestors such that the +++ mutant has the highest $a_{B,2}$. In each case, the mutant successfully displaces its ancestor but reaches a lower final proportion. **c** The population size corresponding to **b** shows that despite the lower proportion of the mutant, the growth rate (proportional to population size) is increasing. **d** An invasion by type 2 cell mutants with successively higher $a_{A,2}$ is shown. The first mutant does not invade, but the second and third do. **e** The population size corresponding to **d** shows the occurrence of paradox 2 during the invasion of the ++ mutant. The population size decreases. Following the successful invasion of the +++ mutant, the population size increases beyond its initial starting point

compared with the situation achieved by the competitive Nash equilibrium, the population grows at a slower rate.

To see how this paradox occurs, we choose a set of production efficiencies and compute the steady-state growth rates of the two cell types as a function of their relative frequencies. Figure 5 shows the growth rates under four scenarios: (1) each cell type maximizing its own growth rate, (2) both types maximizing the growth rate of cells of type 1, (3) both types maximizing the growth rate of cells of type 2, and (4) complete specialization, where each cell type produces only a single metabolite. The population dynamic equilibrium is achieved when the two growth rates are equal. Though the competitive setup leads to a population dynamic equilibrium with a growth rate that is not as large as complete specialization, which happens to be the maximum sustainable growth rate, it comes closer to this

optimum than do the equilibria that result from maximizing the growth of either single cell type.

This paradox demonstrates that if a cell type controls another so as to maximize its immediate growth rate, then, it effectively sacrifices its long-term growth rate. This implies that there is always some long enough time horizon for which this trade-off will result in fewer cells of the controller. Suppose that $dN_i/dt = \zeta g_i N_i$, where $\zeta$ determines the typical population dynamics time scale relative to the typical times over which the chemical concentrations evolve. Then, we can determine how long it takes for the number of type 1 cells in a population following the competitive production dynamics to overtake the number of type 1 cells in a population following the production dynamics controlled by type 1 cells. If the two cell types start at equal frequency, this will happen after a time $t = 13.5\zeta^{-1}$, at which point

the number of type 1 cells in both populations would have grown by a factor of 1420 (Supplementary Fig. 9).

**Appearance of paradoxes in evolving populations.** We have presented three paradoxical behaviors that can occur in simple microbial populations engaged in leaky trade. In each case, we demonstrated the paradox by comparing populations composed of two cell types. Although such comparisons highlight the counterintuitive behaviors, they do not offer any indication as to whether such paradoxes can occur in evolving populations. For example, in the first paradox, we show that a cell type can improve in its efficiency in making a metabolite, and as a consequence, represents a lower proportion of the total population. We did not, however, show that a mutant with a higher efficiency could invade a population, replace its ancestor, and ultimately end up at a lower relative frequency in the population than its ancestor.

To address the issue of evolutionary plausibility, we consider a simple scenario in which a mutant appears in a structured population with two resident cell types (Fig. 6a). The population is composed of different patches, each founded by two cells. Cells grow in patches for a fixed amount of time and then disperse to colonize new, unoccupied patches. The process is repeated until a steady state of relative frequencies of cell types is reached.

Figure 6b shows the results of introducing type 2 cell mutants with a higher $B$ production efficiency in our metapopulation model. We find that each mutant with a higher $a_{B,2}$ is able to invade and replace its ancestor. The steady-state relative frequency of each mutant is less than its ancestor but the total population size—proportional to the community growth rate—increases (see Fig. 6c). For the simulations, type 1 cells are fixed with $a_{A,1} = a_{B,1} = 1$ and all type 2 cells have $a_{A,2} = 0.67$ which are the same parameters as used in Fig. 3. The $a_{B,2}$ values are 2, 4, 10, and 20 for the initial type 2 cell, the + mutant, the ++ mutant, and the +++ mutant, respectively. We chose a growth time in each patch of 12.5 but found that the paradox was observed so long as the time grown in a patch was sufficient to produce 1000 organisms across the population (see Supplementary Note 4). Prior to this time, some mutants cannot invade because there is not sufficient time for growth differences to accumulate and patches with mixed cell types to reach a high enough proportion.

In Fig. 6d, e, we investigate the appearance of paradox 2 by considering invasions of type 2 cell mutants with a higher efficiency of making the $A$ metabolite. Using the same parameters from Fig. 4 where type 1 cells are fixed with $a_{A,1} = a_{B,1} = 1$ and all type 2 cells have $a_{B,2} = 1.82$, we consider an ancestral type 2 cell with $a_{A,2} = 0.2$ and three mutants with $a_{A,2} = 0.57$ (+ mutant), $a_{A,2} = 0.625$ (++ mutant), and $a_{A,2} = 0.71$ (+++ mutant). The first mutant cannot invade because patches with type 1 cells and type 2+ mutants are outgrown by patches with type 1 cells and ancestral type 2 cells. This is indicative of paradox 2 because even though the + mutant is more efficient, it has a lower growth rate when combined with type 1 cells than its less-efficient ancestor. The next mutant can invade because patches with only type 2++ cells outgrow patches with only type 2 cells enough to compensate for any relative losses in mixed populations of type 2++ and type 1 cells. As a result, the entire population size decreases, as predicted by paradox 2. Finally, the +++ mutant is also able to invade but the population size increases. This is a result of an increased growth rate in patches with type 1 and type 2+++ cells. For these simulations, again, we used the time of 12.5. In general, the success of the mutants is much more sensitive to the duration of growth within patches (Supplementary Note 4).

Finally, we consider an evolutionary system involving paradox 3. We assume that a resident population of type 1 and type 2 cells

is invaded by a mutant type 2 cell that can manipulate either cell type's production so as to maximize its own growth. Thus, in mixed patches, whichever cell type is present with the type 2 manipulator cell will have its production altered so as to maximize the growth of the manipulator strain. In this case, we find that the manipulator—despite its advantage—cannot invade the metapopulation from a low relative frequency (see Supplementary Note 5 for the case using parameters from Fig. 5). Even though it benefits at the expense of its partner in mixed populations, these patches are not as productive as other mixed patches without the manipulator. So, patches with type 1 and type 2 cells produce enough cells to overwhelm the manipulator and keep it from invading. If, however, the type 2 manipulator starts its invasion with a high enough relative frequency, it can invade the metapopulation, but in doing so, it drives both type 1 and ancestral type 2 cells extinct (Supplementary Note 5). This leaves the manipulator strain alone in the population with no other cell type to manipulate.

## Discussion

Microbes constantly face decisions about which metabolites to produce. These decisions depend on what metabolites are present in the environment, which, in turn, can be affected by the abundance and production decisions of other microbes. Here, we introduce a simple, general mathematical model to understand the interplay between microbial production decisions and population dynamics. Using this model, we identify the conditions that permit coexistence among different species and discover three paradoxical behaviors that demonstrate the unusual feedback between individual-level production and population-level dynamics.

In our model, beneficial trade emerges naturally as metabolites diffuse in and out of cells and each organism maximizes its own growth rate. We find that different microbes are able to coexist across a broad range of production costs/efficiencies, and in all cases of coexistence, microbes grow faster than if they were isolated from one another. Coexistence occurs only when each of the species are more efficient at producing a different resource. However, this is not a sufficient condition. If one organism is much more efficient at producing one resource and only marginally worse at producing the other, then, it can drive the other species to zero frequency. Thus, there is some threshold for production efficiencies that permits coexistence. In our model, this threshold depends on the growth and production functions of each microbe, as well as the diffusion and metabolite consumption rates. While we investigated the simple case in which each microbe has the same growth function and similar production constraints, in real biological systems, it is likely that these may differ across species, and these differences may result in a richer and more complex set of dynamics[33]. Our set of assumptions concerning overlapping metabolic needs are similar to other models that explore the Black Queen Hypothesis[10, 23] as well as metabolic specialization[34]. The simplicity of the models restricts direct application but helps build intuition about real biological scenarios.

Another consequence of our model is the natural emergence of a division of labor. At each iteration of our model, each microbial species made a production decision that maximized its growth rate in the current environment. Through this simple process, we observed that each microbe shifted its production to the metabolite it is better at producing. Although the microbes did not become complete obligates, we found that this increased specialization led to a higher population growth rate without any external coordination. We note that for some fixed values of production efficiencies/costs, the population as a whole could

grow faster if each microbe completely specialized and this outcome was a stable Nash equilibrium.

The curse of decreased inefficiency provides a mechanism by which the division of labor in mutual obligates (the mutualism category in Table 1) evolves without built-in returns to scale or any benefit of specialization in the production efficiency. In some models of microbial exchange, when one organism loses the ability to make a resource, it grows faster due to a built-in benefit of specialization[10]. This leads to a rapid loss of functionality in coevolving species such that they become mutually reliant on one another. In turn, this loss of functionality can lead to the situation featured in the Black Queen Hypothesis discussed earlier[22]. Our model shows that such increased growth does not require any built-in benefits of specialization. It can simply emerge as a consequence of the fact that a loss in efficiency forces one species to bring its production strategy closer to the globally optimal situation of complete specialization, i.e,. the inverse of paradox 2. Figure 2a shows that the population can grow faster if type 2 cells either increase their efficiency in producing B or decrease their efficiency in producing A.

Another paradox that we uncovered is the curse of increased efficiency, in which as one species becomes more efficient at producing a resource, it becomes rarer in the population. Note that a species that produces both resources less efficiently would experience a similar decline in population. Although these two scenarios exhibit similar qualitative behavior, there are important differences in population structure and stability. In the case of the more efficient species, even though it is rare, it is significant by virtue of its metabolic contribution. If it went extinct due to some stochastic fluctuation, then, the population growth rate would sharply decrease because of the dependency of the more abundant species. In the case of the less-efficient species, the more abundant species does not have any such dependency and would experience little change in its growth rate if the less-efficient species went extinct. These two scenarios may happen in real populations, and without a detailed understanding of the interdependencies that evolved through trade, we may make incorrect inferences about these systems. Indeed, in large microbial consortia, our findings suggest that low-abundance organisms may not necessarily be less fit and, potentially, they could be essential to the fast growth rate of the community. There is evidence of this effect in regard to some microbial communities exposed to antibiotics[35], in which a population of antibiotic-resistant and antibiotic-sensitive types depends on the efficiency with which resistant types break down the antibiotics, and decreasing the efficiency of resistant types increases their relative frequency in the population.

The paradox of increased efficiency is related to the trade-off between growth and relative abundance presented in ref. [6]. In both cases, modification of a species' trait decreases its relative abundance in the population but yields a faster-growing population. In ref. [6], this trait is the extent to which a species exports a particular metabolite into the environment. A species can increase its relative abundance by restricting the export of metabolites, but the population as a whole grows more slowly. In the absence of selection for faster population growth, species will tend to evolve the restricted export of metabolites[6]. In contrast, the modifiable trait in our model is the efficiency of producing a particular metabolite. As a species increases its efficiency, it decreases in relative abundance but the population grows faster. If a single organism of the species were to decrease its efficiency, it would be outcompeted. If, instead, the single organism were to increase its efficiency, it can successfully invade. Thus, there is selection to increase the metabolic efficiency of the species despite the trade-off in relative abundance.

The third paradox that we uncovered, the curse of control, shows how exploitation of one species by another can lead the whole system to a worse growth rate. This paradox demonstrates a problem with forms of parasitism and cheating via chemical manipulation[31, 36]. It is a stronger form of exploitation than cheating via foregoing the production of a costly public good[21], because one species completely controls another's production and manipulates it to maximize its own growth. The short-term gains in population abundance that arise by these strategies lead to long-term losses in the population growth rate. Depending on the length of time of an association, it may be more beneficial to compete with another microbe than to exploit it. In our meta-population model, we found that even if a cell type can manipulate all others so as to maximize its own growth rate, it can fail to invade the population because of the greater production of competing populations without the manipulator. When the manipulator did invade the population, it drove all other cell types extinct, rendering its ability to manipulate useless.

In this paper, we restricted ourselves to the study of a particularly simple case of a much more general model. Though we consider only two metabolites and two cell types with equal requirements for growth, we not only illustrate the uncoordinated emergence of beneficial trade within a coexisting population, but also uncover a rich landscape of unexpected outcomes. Our model can be generalized to study the interaction of any number of cell types, exchanging any number of valuable molecular species, with arbitrary growth rates given as functions of the concentrations of these molecules, and with arbitrary constraints on their production rates. We expect that by adding more complexity to our model, we will be able to model a large range of emergent behavior that may be present in a real microbial community and may run counter to common intuition and implicit assumptions about the driving principles in these communities.

## Methods

**Model description**. We consider a microbial population model in which organisms require two metabolites $A$ and $B$ in order to grow and reproduce. We assume that the population growth rate of cell type $i$ is proportional to a rate $g_i(A_i, B_i)$ determined by its internal concentration of the metabolites. Although there may be many possible growth functions $g_i$, we choose the general functional form $g_i(A_i, B_i) = k_i A_i B_i$. This represents a mass action law for an elementary reaction, wherein $A$ and $B$ react to form a product that is used directly for growth. We consider the simple case of the growth functions $g_i$ in which both organisms have the same growth function and $k_i = 1$. This assumption implies that the organisms have similar metabolic needs. As a consequence of the growth process, metabolites $A$ and $B$ are consumed at rates $s_{A,i}g_i(A_i, B_i)$ and $s_{B,i}g_i(A_i, B_i)$, respectively. The stoichiometry coefficients, $s_{A,i}$ and $s_{B,i}$, depend on the growth reaction and here, we investigate the simple case where $s_{A,i} = s_{B,i} = 1$.

We assume that each cell type $i$ can produce both metabolites, but that the production rates of $A$ and $B$ metabolites, denoted by $p_{A,i}$ and $p_{B,i}$, are subject to a budget constraint that reflects a finite amount of resources that can be devoted toward production. We encapsulate the relevant constraints in the production constraint function, $P_i(p_{A,i}, p_{B,i})$, subject to a constraint $P_i(p_{A,i}, p_{B,i}) \leq P_{max}$ where $p_{X,i}$ is the rate of production of metabolite $X$ by cells of type $i$. For example,

$$P_i(p_{A,i}, p_{B,i}) = c_{A,i}p_{A,i} + c_{B,i}p_{B,i} \leq 1 \qquad (1)$$

represents a situation where metabolites $A$ and $B$ can be produced at fixed costs ($c_{A,i}$ and $c_{B,i}$, given in units of the total budget) independent of the total rate of production. Thus, there are no returns to scale.

In addition to production and consumption, we assume that there is a rate of passive diffusion of metabolite molecules out of any cell and into a random other cell. The total flux of molecules leaving cells of any type will be proportional to a diffusion coefficient $D$, the intracellular concentrations of the molecules, and the number of cells of this type. Since diffusion is unbiased in our model, molecules will enter cells of type 1 or 2 according to their proportions in the population. We define the relative frequency of cells of type $i$ by $n_i = N_i/(N_1 + N_2)$, where $N_i$ is the number of cells of type $i$. As a result, the net flux of $A$ molecules entering a single cell of type 1 is $Dn_2(A_2 - A_1)$ and similarly $Dn_1(A_1 - A_2)$ for cells of type 2. The diffusion coefficient $D$ determines the relative rate at which molecules flow down a gradient as opposed to getting consumed by the growth reaction. Therefore, the smaller $D$ is, the more benefit a microbe derives from producing a metabolite directly as opposed to relying on a trading partner. We use $D = 3$ in the numerical

cases investigated in the main text of the paper, but show the effects of varying $D$ in the Supplementary Material (Supplementary Note 1).

Besides cross-cell diffusion, we assume that there is a rate $\mu$ at which metabolites are lost and not regained by any cell. We set this loss rate to be $\mu = 0.05$ throughout the paper. Although the precise value of $\mu$ does not change the key results of the paper, we analyze the effects of varying $\mu$ in the Supplementary Material (Supplementary Note 1).

These dynamical processes result in a set of differential equations that describes the intracellular concentrations of $A$ and $B$ metabolites in the two types of cells:

$$
\begin{aligned}
\frac{dA_1}{dt} &= p_{A,1} + Dn_2(A_2 - A_1) - \mu A_1 - s_{A,1}g_1(A_1, B_1), \\
\frac{dB_1}{dt} &= p_{B,1} + Dn_2(B_2 - B_1) - \mu B_1 - s_{B,1}g_1(A_1, B_1), \\
\frac{dA_2}{dt} &= p_{A,2} + Dn_1(A_1 - A_2) - \mu A_2 - s_{A,2}g_2(A_2, B_2), \\
\frac{dB_2}{dt} &= p_{B,2} + Dn_1(B_1 - B_2) - \mu B_2 - s_{B,2}g_2(A_2, B_2).
\end{aligned}
\tag{2}
$$

This dynamical system is similar to the one Taillefumier et al.[34] used to study coordination among bacterial populations when exposed to a diverse resource supply. In our situation, we have no externally supplied metabolites and do not allow cells to interconvert metabolites: all metabolites are immutable and produced by the cells themselves. We also simplify the system by modeling diffusion between cells rather than explicitly modeling the extracellular environment. As a result, our system has three types of coevolving dynamic variables: the intracellular metabolite concentrations ($A_i$ and $B_i$), the production terms ($p_{A,i}$ and $p_{B,i}$), and the relative population frequencies of the cell types ($n_1$ and $n_2$).

We assume that the dynamics with which the population sizes, $N_1(t)$ and $N_2(t)$, evolve is much slower than the rates of metabolite production and diffusion. At shorter time scales, the metabolite concentrations reach a steady state, where the time derivatives on the left-hand sides of Eq. (2) equal zero. For particular values of the production rates, the steady-state values of the growth functions, denoted as $g_1^*$ and $g_2^*$, can be determined by solving the resulting algebraic equations. If $g_1^* > g_2^*$, then, $n_1$, the relative frequency of cells of type 1, grows, thereby altering Eq. (2). Since an increased $n_1$ affects the values of the steady-state growth rates, we then resolve for the steady state with the increased $n_1$. This iterative process continues until a stable population equilibrium is reached. In order for the system to be in a stable equilibrium with both cell types at nonzero frequency, the steady-state growth rates must be equal, that is, $g_1^* = g_2^*$. Alternatively, there could be an equilibrium where one cell type has a higher growth rate, while the other cell type is driven to a relative frequency approaching zero.

We have not yet discussed how the production rates are chosen subject to the budget constraint. In our model, we assume that the production rates, $p_{X,i}$, can vary in time. One possibility of how this happens is to assume that the organisms can regulate their production rates on a fairly short time scale, and cells of each type adjust their own production rates so as to maximize their growth, subject to the perceived external conditions. This assumption leads to a situation where each cell type's choice of production rates is the best response to the external conditions, which are the result of the choice of the other cell type, implying a Nash equilibrium. While the assumption that the production can be regulated on a shorter time scale than the population dynamic time scale that is convenient, it is not necessary even if regulation only occurs through mutations, the system will be driven to a Nash equilibrium by the fact that a population not using the best-response production rates is susceptible to invasion by a mutation that does (Supplementary Note 2). Also, see Supplementary Note 3 for a discussion of changing functional forms of the model.

**Code availability**. A computer code for our analyses is provided as Supplementary Files.

**Data availability**. No data were produced in this study.

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

## Acknowledgements

We acknowledge helpful discussions with David Wolpert. Y.K. and E.L. acknowledge the support of the Santa Fe Institute through the Omidyar Fellowship.

## Author contributions

Y.K., J.H.M. and E.L. designed the study. Y.K. and E.L. performed the analyses. Y.K., J.H.M. and E.L. wrote the paper.

## Additional information

**Competing interests:** The authors declare no competing financial interests.

