## [Peer Review file · Nature Communications]

Reviewers' comments:

Reviewer #1 (Remarks to the Author):

This paper develops theory to study how two microbial strains, which each can produce the same two costly secreted factors, coevolve in response to different changes in parameter values. Each strain is better at producing one of the factors by construction and the result is that the two often evolve to specialize (as expected). However, the authors then show that, once this specialization has arisen, how manipulating efficiency of metabolite production can lead to initially counter-intuitive outcomes. For example, becoming more efficient at making one of the two products can lead to a strain making both. This loss of specialization can make the system less efficient over all because the system is set up such that specialization leads to most productivity, such that the focal strain ultimately loses out. The authors then consider how giving one strain control over the mutualism affects the outcome and find that this leads to the strain increasing in frequency but the whole system reducing in productivity. In general, I like the fact that the authors are looking into eco-evolutionary dynamics and coevolution which are interesting and important new areas for microbiology. However, the specific scenario studied and the modelling approach has major limitations that make the generality of the results questionable.

The key concern is that the authors are looking at coevolution between two strains that never disperse or interact with other strains. The staggering diversity of microbial systems means that this strong coevolution scenario is extremely restrictive, and in an important way. The evolution of strict dependence between microbial strains, which is what the whole paper rests upon, is fragile to the assumption that strains do not always interact with the same coevolving strain. More specifically, there is a model with a very similar starting premise (Oliveira et al. 2014 PNAS) that predicts that evolution of such strict dependence will often be disfavored. This paper is not cited.

But let's assume that two microbial strains have somehow found themselves next to each other more or less indefinitely, would the effects shown in the model evolve? Not necessarily, the model I think also requires that both strains make two products that both can use, and that one strain is better at producing each. We are looking then at a specialized and specific condition of reciprocal coevolution between two microbial strains that can each offer the other something, which may sometimes occur but it is far from clear that the effects discussed are common within microbial communities.

More specifically, while initially counter intuitive, all of the results become relatively intuitive when one realizes that the authors have created an ecology where two strains have divided labor and are strongly dependent on one another. Once strains are so dependent, anything that throws this productive specialization off, can feedback with costly effects. This is an interesting point for specialized mutualisms perhaps so rephrasing the paper in terms of weird effects of unusually strong co-dependence in microbial mutualisms might be a legitimate way to frame this? But the fact that mutualism leads to dependence and that this leads to system fragility and counter intuitive dynamics is a point that has been made in theoretical ecology and evolution.

A final point is that the outcome of the control paradox I found the least surprising. We know well that species will evolve traits that improve their evolutionary fitness at the expense of their population size. Ad absurdum, this is called evolutionary suicide (see Rankin and López-Sepulcre 2005 *Oikos*), but a simple example in microbes is the evolution of antibiotic production to inhibit other strains. Antibiotic production can be favoured even if it reduces a strains cell numbers, so long as it reduces the other strains cell numbers even more.

In sum, the authors are working in an interesting area and I hope they will continue to do so, because there is a lot of potential in their approach and goals. However, I also encourage them to rethink their core assumptions and approach to better reflect the likely ecology of microbial communities. As it stands, the current model has greatly restricted the applicability of their work

and conclusions, and my impression is that more attention to published work in ecology and evolution of mutualisms and microbes will serve them well.

Reviewer #2 (Remarks to the Author):

In "Paradoxes in Leaky Microbial Trade", the authors explore a simple model of metabolic exchange and are able to extract a number of interesting conceptual insights that are likely general across models and in real microbial interactions. Overall I found the model reasonable, the conclusions surprising and relevant to many microbial communities, and the writing clear and compelling. My only concern is whether the qualitative conclusions would change if the modeling assumptions were altered. Below I elaborate on the growth function and the production constraint.

The growth rate function (Eq 1) is reasonable, but left me wondering whether any of the qualitative conclusions would have been different with a more general Monod growth equation where $g = (A/(A+K))(B/(B+K))$. The nice thing about this form is that it captures the basic features of the growth function (in particular saturation), so that growth rate doesn't go to infinity if one of the resources goes to infinity.

The production constraint function (Eq 2) is broadly reasonable, but is again not the most obvious choice to use so then left me wondering how important it is for the qualitative conclusions. In particular, most authors simply assume a cost per unit of making each resource, where this cost decreases the growth rate directly. This assumption is simple and has empirical support (eg the work from Terry Hwa's lab, Scott et al, Science (2010)).

The model used by the authors does not explicitly consider the metabolites A and B being present in the media, and instead assumes that the metabolites directly diffuse from one cell to another. Do the authors know how this assumption influences their results?

line 181: If the two strains had a different base growth rate would it be possible to observe a stable equilibrium in which both strains benefit? In various models that we have played with we did not see this, but I think that it would be interesting.

Paradox 1: This is nice, and is likely a phenomenon that is observed in a range of models and systems. For example, we observed something similar in experiments between a cooperator and cheater in which the cooperator is a resistant cell that breaks down an antibiotic and the cheater is a sensitive cell that doesn't break down the antibiotic. In this case, adding an inhibitor that decreased the rate of antibiotic breakdown (and in principle directly harming the resistant strain) had the effect of increasing the resistant fraction at equilibrium (Yurtsev et al, MSB (2013)).

Despite not being surprised by paradox #1, I was still surprised by paradox #2. Very interesting.

I don't understand paradox #3. If strain 1 can control both production rates then doesn't it by definition have access to the maximum possible sustainable growth rate? I wrote the previous two sentences before getting to the final paragraph of this section. I am fine with the statement that local optimization is not sufficient to reach the global optimum, but this should be stated clearly in the initial definition of the paradox.

Minor points:

I normally catch a dozen small errors when reviewing a manuscript, but I didn't notice any here. Very well written and carefully prepared.

I found this paper to be a pleasure to read.

Jeff Gore

Reviewer #3 (Remarks to the Author):

The manuscript investigates an exciting and important topic, which is the interplay between microbial metabolite production decisions and population dynamics. This is likely an area that would be of an increasing interest over the coming years. The authors imagine a particular scenario in which two bacteria have different comparative advantages – they have different unit costs of metabolite production. They also imagine that the microbes can adjust the production of two metabolites that are required for growth so that they maximize their growth rates. Perhaps not too surprisingly, the authors show that the comparative advantage can lead to coexistence and division of labor when the two types of microbes are allowed to trade two metabolites. Then they give three examples of counterintuitive phenomena that can arise in such a system. Two of them say that if the unit cost of production of a metabolite goes down (for some reason) then we don't necessarily see increase of both growth rate and relative frequency. We can see increase in only one and not the other. The third paradox is that competing with another microbe might lead to a higher eventual population growth rate than controlling the other microbe. I am overall supportive of this study but have several concerns and suggestions:

1) The authors claim that whether production rates are optimized instantaneously to achieve maximal growth (though some magic mechanism) or whether they are hardwired and evolve through mutations doesn't make a difference for the outcome. Some simulation results are presented in the supplementary to support this. First of all, the statement that regulation on small time scales and evolution have the same effect is interesting and, if true, can be elevated to one of the main results, rather than something that is part of the model section. It is a result that would highlight the generality of the phenomena investigated. Ideally, the authors should offer a mathematical proof that the mode of adjustment does not matter. Alternatively, proper evolutionary simulations can be done, say in which mutants are introduced one by one and the dynamics is equilibrated in-between. Mutations that change production levels by large amounts can be used to prevent the dynamics from being stuck in local optima.

2) Isn't there supposed to be a term diluting internal concentrations in proportion to growth rate in eq.3? Would such a term change the dynamics and conclusions? Or are the results presented only valid in the limit when growth rate is negligible compared to loss rate? Sometimes such terms lead to multi-stability of the dynamics.

3) The model is introduced only with equations, and, even then, some of the dynamics is kept implicit (for example the population dynamics). It would help a lot to spell out a biological realization corresponding to the equations. For example, would bacteria growing in a chemostat result in exactly the equations used? Is that the situation the authors imagine? This would help a lot with making the paper accessible to a broad readership. But even for mathematically sophisticated readers, it is a bit hard to grasp what the biological situation is. For example, reverse engineering the logic of their diffusion term, I conclude that there is a container of a finite size, which would permit fast equilibration rather than metabolites leaking away to infinity. But what happens to that container as the bacteria exponentially fill it up? Such things need to be transparent to the reader. Fig 1 can be a diagram of the biological situation(s) considered. It would also be helpful to provide a fully explicit dynamic model in the supplementary and then show how it is reduced and what are the assumptions.

4) Paradox 3 is an interesting theoretical construct but is quite artificial for microbes. Are there

any plausible mechanisms (one can imagine) for microbes to completely manipulate the production levels of other microbes? The beauty of the coexistence and division of labor that the authors describe is that it arises simply and naturally (assuming point 1) is fully addressed). Here we don't have that. I can understand the difference between competition and manipulation if the production rates are instantaneously tuned to maximize growth rates and then population dynamics acts as a slow degree of freedom. But when I think about the situation when production rates are adjusted evolutionary it gets confusing. If you have two mutants (alleles) that try to manipulate the production of the other microbes differently, then what happens? Each of them manipulates a fraction of the population, or are the manipulating influences averaged somehow? What is the paradox in evolutionary terms: that the manipulator strategy would lose against the one that cannot manipulate? Finally, even if one accepts the setup and result, Fig 5. does not really help to understand the final outcome because these lines are plotted for particular production rates. So, unlike the other paradoxes the why question is not satisfactorily answered, which is OK but the authors should either try to explain better the intuition for why we see the outcome or remove it. Overall, I think paradox 3 is problematic on several levels, and it might be easier for the authors to remove it, save it for another paper, or put it in the supplementary as a curiosity which might not be relevant for microbes but connects to some branches of game theory.

5) For the other two paradoxes, it would be nice to phrase them in evolutionary terms. Are the authors saying that a higher efficiency mutant would be selected against? If yes, this would be a way to strengthen the paradoxes. If not, then we go back to some of the issues in 1).

REVIEWER 1 COMMENTS

This paper develops theory to study how two microbial strains, which each can produce the same two costly secreted factors, coevolve in response to different changes in parameter values. Each strain is better at producing one of the factors by construction and the result is that the two often evolve to specialize (as expected). However, the authors then show that, once this specialization has arisen, how manipulating efficiency of metabolite production can lead to initially counter-intuitive outcomes. For example, becoming more efficient at making one of the two products can lead to a strain making both. This loss of specialization can make the system less efficient over all because the system is set up such that specialization leads to most productivity, such that the focal strain ultimately loses out. The authors then consider how giving one strain control over the mutualism affects the outcome and find that this leads to the strain increasing in frequency but the whole system reducing in productivity. In general, I like the fact that the

authors are looking into eco-evolutionary dynamics and coevolution which are interesting and important new areas for microbiology. However, the specific scenario studied and the modelling approach has major limitations that make the generality of the results questionable.

Response: Thank you for your constructive comments. We have made a significant attempt to demonstrate the wider applicability of our modeling results, including the addition of a meta-population model consisting of patches and dispersal.

The key concern is that the authors are looking at coevolution between two strains that never disperse or interact with other strains. The staggering diversity of microbial systems means that this strong coevolution scenario is extremely restrictive, and in an important way. The evolution of strict dependence between microbial strains, which is what the whole paper rests upon, is fragile to the assumption that strains do not always interact with the same coevolving strain. More specifically, there is a model with a very similar starting premise (Oliveira et al. 2014 PNAS) that predicts that evolution of such strict dependence will often be disfavored. This paper is not cited.

Response: We now include a meta-population model in a new section of the paper. In this model strains can interact and disperse, similar to the model of Oliveira et al. 2014 PNAS. We show that the paradoxical behaviors we identify in our model with two cells types can also evolve in meta-populations with a type 1 cell strain, a type 2 cell strain, and different type 2 cell mutant strains. We believe that this addresses the reviewer's concern.

We also want to clarify that in our model the cell types do not become strictly dependent on one another. Both cell types always retain the ability to make the two metabolites needed for growth. In this way, the scenario is different from Oliveira et al. 2014 PNAS. The meta-population model highlights this difference in that a clonal population can still reproduce. Thus, a population of all type 1 cells can still successfully reproduce. The fact that there is not strict dependence can be helpful in the evolutionary appearances of our paradoxes. Consider paradox 2 where a type 2 cell mutant strain has higher efficiency at making the *A* metabolite but lower yield when coupled with type 1 cells. If type 2 cells could only exist in the presence of type 1 cells then we would not expect this mutant to invade. Instead, it can invade because although the growth rate of type 1-type 2 mutant communities is lower than type 1-type 2 ancestor communities, the growth rate of clonal type 2 mutant communities is higher than clonal type 2 ancestor communities. Thus, the plasticity of production in our model facilitates the evolutionary appearance of paradox 2.

But lets assume that two microbial strains have somehow found themselves next to each other more or less indefinitely, would the effects shown in the model evolve? Not necessarily, the model I think also requires that both strains make two products that both can use, and that one strain is better at

producing each. We are looking then at a specialized and specific condition of reciprocal coevolution between two microbial strains that can each offer the other something, which may sometimes occur but it is far from clear that the effects discussed are common within microbial communities.

Response: We agree that there are some specific constraints on our model. It is true that if one strain is absolutely better at making both compounds than another then it will out-grow the other strain. There is obviously the potential for co-evolutionary specialization but this is outside the scope of our paper. It is also true that our model assumes that both strains need the same resources to survive. The generality of this assumption might seem limiting but it is at the heart of many other cross-feeding/metabolic network/Black Queen papers (including Oliveira et al. 2014 PNAS). There may be many organisms which require different resources but we believe that there are also strong overlaps as many organisms must produce the same types of compounds to survive (amino acids, enzymes, etc.). There is also some evidence that these assumptions might be satisfied in some wild microbial communities such as marine photosynthetic organisms or maybe even some microbiomes though the evidence there is certainly not definitive. Outside of natural microbial communities, where much is unknown, our results might have interesting applications in synthetic communities. Indeed, our results suggest that if there is strong selection for growth or biomass production from a two strain community then the decreasing relative frequency of one strain does not necessarily mean it is less fit, as it could be necessary to the enhanced production of the entire community.

All of this is to say that we understand the reviewer’s concerns. To address these issues, we have added more discussion of the applicability of our model and its assumptions. We also include a supplementary section in which we explore different functional forms in the model and show their effects.

More specifically, while initially counter intuitive, all of the results become relatively intuitive when one realizes that the authors have created an ecology where two strains have divided labor and are strongly dependent on one another. Once strains are so dependent, anything that throws this productive specialization off, can feedback with costly effects. This is an interesting point for specialized mutualisms perhaps so rephrasing the paper in terms of weird effects of unusually strong co-dependence in microbial mutualisms might be a legitimate way to frame this? But the fact that mutualism leads to dependence and that this leads to system fragility and counter intuitive dynamics is a point that has been made in theoretical ecology and evolution.

Response: The phrasing of “strongly dependent” is a bit unclear. In terms of our model (and now highlighted in the meta-population model) the organisms are not strictly dependent on each other. At any given time, each organism can survive in isolation. However, when they share the same environment, the combination of differential production and

leakiness allows for emergent coordination and division of labor. This occurs simply as a result of each cell choosing production so as to maximize its own fitness. We should also say that complete division of labor is not a certainty. There is a large area of parameter space shown in Figure 2b in which at least one cell type is making both goods. This depends on the diffusion rate which affects the privatization of the metabolites. We would prefer not to reframe the paper in terms of mutualisms because we think that it would confuse readers given the metabolic plasticity of the two cell types. We are considering a stage prior to an obligate mutualism.

Another example of the lack of strict dependence is paradox 3's treatment within our meta-population model. In a population of type 1 cells, type 2 cells can successfully invade because type 1-type 2 communities are more productive than type 1 cells alone. Furthermore, type 1 and type 2 cells coexist for much of the parameter space considered. In contrast, type 2 manipulator cells which have the same metabolic efficiencies as type 2 cells cannot invade unless they start with high relative frequencies. When they do invade, there is no coexistence between type 2 manipulator cells and any other type of cell. So despite the same metabolic efficiencies of the type 2 cells and the manipulators, we observe different potentials for invasion and coexistence.

A final point is that the outcome of the control paradox I found the least surprising. We know well that species will evolve traits that improve their evolutionary fitness at the expense of their population size. Ad absurdum, this is called evolutionary suicide (see Rankin and Lopez-Sepulcre 2005 Oikos), but a simple example in microbes is the evolution of antibiotic production to inhibit other strains. Antibiotic production can be favoured even if it reduces a strains cell numbers, so long as it reduces the other strains cell numbers even more.

Response: It is true that organisms can evolve apparently costly traits that can be beneficial. However, in the example we consider the organisms are evolving apparently beneficial traits that are costly. Moreover, there is a new result concerning paradox 3 within the meta-population model. If we assume that there is a type 2 manipulator strain than can manipulate both type 1 and type 2 cells for its own benefit—i.e. when paired with either type it increases in relative frequency and absolute number—then we might believe that such a manipulator could easily invade the population. However, we find that it cannot invade from low frequency primarily because the type 1-type 1 populations and type 1-type 2 populations are sufficiently productive to outgrow it. This is an example of a form of parasitism that cannot successfully invade. In addition, we find that at high frequency it can invade but cannot coexist with any other cell type. We believe this is qualitatively different from the examples provided by the referee.

In sum, the authors are working in an interesting area and I hope they will continue to do so, because there is a lot of potential in their approach and goals. However, I also encourage them to rethink their core assumptions

and approach to better reflect the likely ecology of microbial communities. As it stands, the current model has greatly restricted the applicability of their work and conclusions, and my impression is that more attention to published work in ecology and evolution of mutualisms and microbes will serve them well.

Response: Thank you for your comments. We believe we have addressed Reviewer 1’s three main concerns by: 1) adding a meta-population model that demonstrates that the paradoxes we found can arise via the successful invasion of mutants, 2) clarifying the issue that our cell types are not strictly dependent on one another, and 3) adding more analysis and discussion of the applicability of our model’s assumptions, and tying these to the large body of existing literature.

REVIEWER 2 COMMENTS

In “Paradoxes in Leaky Microbial Trade”, the authors explore a simple model of metabolic exchange and are able to extract a number of interesting conceptual insights that are likely general across models and in real microbial interactions. Overall I found the model reasonable, the conclusions surprising and relevant to many microbial communities, and the writing clear and compelling. My only concern is whether the qualitative conclusions would change if the modeling assumptions were altered. Below I elaborate on the growth function and the production constraint.

Response: Thanks for that excellent summary and positive feedback.

The growth rate function (Eq 1) is reasonable, but left me wondering whether any of the qualitative conclusions would have been different with a more general Monod growth equation where $g = (A/(A+K))(B/(B+K))$. The nice thing about this form is that it captures the basic features of the growth function (in particular saturation), so that growth rate doesn’t go to infinity if one of the resources goes to infinity.

Response: We now include a supplementary section exploring the effects of using Monod growth kinetics. In general we find the same qualitative paradoxical behaviors. One interesting difference is that when type 2 cells have very high B metabolite efficiency, there appear to be multiple Nash equilibria such that bistability is possible.

The production constraint function (Eq 2) is broadly reasonable, but is again not the most obvious choice to use so then left me wondering how important it is for the qualitative conclusions. In particular, most authors simply assume a cost per unit of making each resource, where this cost decreases the growth rate directly. This assumption is simple and has empirical support (eg the work from Terry Hwas lab, Scott et al, Science (2010)).

Response: We also include an interaction between the production and a cost for growth and discuss how this assumption affects our results in a new supplementary section. In short, it depends on whether there is a linear or quadratic soft constraint.

The model used by the authors does not explicitly consider the metabolites A and B being present in the media, and instead assumes that the metabolites directly diffuse from one cell to another. Do the authors know how this assumption influences their results?

Response: The model we use has similarities to a model by Taillefumier *et al* that considers an inflow of nutrients and explicitly models the external concentrations of metabolites in the medium. In our model, because there is no inflow of metabolites, we found that treating the external concentrations implicitly simplified the analytical considerations, and was warranted when considering steady states of the reaction-diffusion kinetics. That said, we now include in the supplementary material a derivation of the simplification from the model that treats the external concentration explicitly to the model used in the main text.

line 181: If the two strains had a different base growth rate would it be possible to observe a stable equilibrium in which both strains benefit? In various models that we have played with we did not see this, but I think that it would be interesting.

Response: We also include an analysis in which the organisms have different base growth rates in the supplementary section. As the referee suggests, we do find cases in which at least one of the cell types does better in isolation than when trading.

Paradox 1: This is nice, and is likely a phenomenon that is observed in a range of models and systems. For example, we observed something similar in experiments between a cooperator and cheater in which the cooperator is a resistant cell that breaks down an antibiotic and the cheater is a sensitive cell that doesn't break down the antibiotic. In this case, adding an inhibitor that decreased the rate of antibiotic breakdown (and in principle directly harming the resistant strain) had the effect of increasing the resistant fraction at equilibrium (Yurtsev et al, MSB (2013)).

Response: Thank you for this example. It is very interesting and certainly relevant. We now include it in our Discussion.

Despite not being surprised by paradox #1, I was still surprised by paradox #2. Very interesting.

Response: Great!

I dont understand paradox #3. If strain 1 can control both production rates then doesnt it by definition have access to the maximum possible sustainable growth rate? I wrote the previous two sentences before getting to the final paragraph of this section. I am fine with the statement that local optimization is not sufficient to reach the global optimum, but this should be stated clearly in the initial definition of the paradox.

Response: Thanks—we think when it is framed in terms of local and global optima it makes intuitive sense.

I normally catch a dozen small errors when reviewing a manuscript, but I didnt notice any here. Very well written and carefully prepared. I found this paper to be a pleasure to read.

Response: Thanks so much.

REVIEWER 3 COMMENTS

The manuscript investigates an exciting and important topic, which is the interplay between microbial metabolite production decisions and population dynamics. This is likely an area that would be of an increasing interest over the coming years. The authors imagine a particular scenario in which two bacteria have different comparative advantages they have different unit costs of metabolite production. They also imagine that the microbes can adjust the production of two metabolites that are required for growth so that they maximize their growth rates. Perhaps not too surprisingly, the authors show that the comparative advantage can lead to coexistence and division of labor when the two types of microbes are allowed to trade two metabolites. Then they give three examples of counterintuitive phenomena that can arise in such a system. Two of them say that if the unit cost of production of a metabolite goes down (for some reason) then we dont necessarily see increase of both growth rate and relative frequency. We can see increase in only one and not the other. The third paradox is that competing with another microbe might lead to a higher eventual population growth rate than controlling the other microbe. I am overall supportive of this study but have several concerns and suggestions:

Response: Thank you for the summary of our findings and positive comments.

1) The authors claim that whether production rates are optimized instantaneously to achieve maximal growth (though some magic mechanism) or whether they are hardwired and evolve through mutations doesnt make a difference for the outcome. Some simulation results are presented in the supplementary to support this. First of all, the statement that regulation on

small time scales and evolution have the same effect is interesting and, if true, can be elevated to one of the main results, rather than something that is part of the model section. It is a result that would highlight the generality of the phenomena investigated. Ideally, the authors should offer a mathematical proof that the mode of adjustment does not matter. Alternatively, proper evolutionary simulations can be done, say in which mutants are introduced one by one and the dynamics is equilibrated in-between. Mutations that change production levels by large amounts can be used to prevent the dynamics from being stuck in local optima.

Response: We have expanded our section on this in the supplementary material. The reason we assume that the time scale of production is fast compared to reproduction is that many organisms regulate their metabolism in such a way so as to maximize growth. This is at the heart of metabolic network models using FBA which have had some success in predicting evolutionary population dynamics. While we agree that a mathematical proof would be ideal, we found it much more straightforward to simply include evolutionary simulations to support our claim. As suggested by the reviewer, we also introduce mutants one by one and the dynamics is equilibrated in-between.

2) Isn't there supposed to be a term diluting internal concentrations in proportion to growth rate in eq.3? Would such a term change the dynamics and conclusions? Or are the results presented only valid in the limit when growth rate is negligible compared to loss rate? Sometimes such terms lead to multi-stability of the dynamics.

Response: We are not exactly sure what is meant by this comment. The internal concentrations of metabolites are decreased as a consequence of population growth. This occurs because of our mass-action assumption that growth rate is proportional to kAB . Perhaps the reviewer is asking a different question as to whether when an individual cell grows and its volume changes that this effectively reduces its internal concentration of metabolites? If so, then we acknowledge that our approach does not consider such a fine-grained perspective. Instead, we are using a population average perspective that could be interpreted as the average response across a cell type.

Interestingly, in our first models of this system we did consider the diluting effects of cell volume changes so that when cells increase in volume this decreases the internal concentrations of metabolites. However, this was compensated by a decreased growth and an increase in the diffusion-based flux into the cell. Thus, it did not seem to have a significant effect on the population dynamics. One issue with the volume-based models, though, was that in a deterministic set of differential equations they imply synchronous reproduction across each cell type. To correct for this, one could consider stochastic equations, different lineages of synchronized cell types, or use the population-level approach we adopted. Ultimately we chose our current model because it has the same key features as the others with fewer parameters and it is consistent with models used in other studies of metabolic

cross-feeding. This modeling choice is consistent with, and indeed can be thought of as a consequence of, other choices we have made, such as that the population dynamics proceeds much more slowly than the reaction-diffusion dynamics.

3) The model is introduced only with equations, and, even then, some of the dynamics is kept implicit (for example the population dynamics). It would help a lot to spell out a biological realization corresponding to the equations. For example, would bacteria growing in a chemostat result in exactly the equations used? Is that the situation the authors imagine? This would help a lot with making the paper accessible to a broad readership. But even for mathematically sophisticated readers, it is a bit hard to grasp what the biological situation is. For example, reverse engineering the logic of their diffusion term, I conclude that there is a container of a finite size, which would permit fast equilibration rather than metabolites leaking away to infinity. But what happens to that container as the bacteria exponentially fill it up? Such things need to be transparent to the reader. Fig 1 can be a diagram of the biological situation(s) considered. It would also be helpful to provide a fully explicit dynamic model in the supplementary and then show how it is reduced and what are the assumptions.

Response: Thanks for raising this issue. We now include a more explicit model in the supplementary section that highlights the scenarios we are considering. The biological interpretation could be patches, ponds, or enclosed areas such as flasks. This is similar to other such models in the field of cross-feeding. To elucidate this, we now include a schematic in our new meta-population model where organisms grow for a finite time in separate patches and then disperse to new patches (similar to the model of Oliveira et al in PNAS 2014).

4) Paradox 3 is an interesting theoretical construct but is quite artificial for microbes. Are there any plausible mechanisms (one can imagine) for microbes to completely manipulate the production levels of other microbes? The beauty of the coexistence and division of labor that the authors describe is that it arises simply and naturally (assuming point 1) is fully addressed). Here we don't have that. I can understand the difference between competition and manipulation if the production rates are instantaneously tuned to maximize growth rates and then population dynamics acts as a slow degree of freedom. But when I think about the situation when production rates are adjusted evolutionary it gets confusing. If you have two mutants (alleles) that try to manipulate the production of the other microbes differently, then what happens? Each of them manipulates a fraction of the population, or are the manipulating influences averaged somehow? What is the paradox in evolutionary terms: that the manipulator strategy would lose against the one that cannot manipulate? Finally, even if one accepts the setup and

result, Fig 5. does not really help to understand the final outcome because these lines are plotted for particular production rates. So, unlike the other paradoxes the why question is not satisfactory answered, which is OK but the authors should either try to explain better the intuition for why we see the outcome or remove it. Overall, I think paradox 3 is problematic on several levels, and it might be easier for the authors to remove it, save it for another paper, or put it in the supplementary as a curiosity which might not be relevant for microbes but connects to some branches of game theory.

Response: We agree that paradox 3 is somewhat of an outlier when compared to paradoxes 1 and 2. We would prefer to leave it in the manuscript if possible because we think it might be of interest to some audiences—especially now that we have some new results concerning the inability of a manipulator to coexist with other types. We include this paradox because we were inspired by the manipulations of hosts by parasites. In order to spread to new hosts or obtain more resources, parasites manipulate host behavior (*Toxoplasma gondii* comes to mind). In the microbe case, we agree that we did come up with a theoretical construct to explore what might happen if one type of organism could instantaneously manipulate another. There is evidence for the manipulation of microbe behavior via external molecules, e.g. quorum-sensing and AI-2. Though in our particular case, we do not invoke any molecular mechanism.

We find that manipulation leads to a community with a slower growth rate than when the population was competing. This effect prevents the manipulator from invading at low frequencies. The multiple manipulator scenario is beyond our scope here. We agree that there could be issues if different strains have the ability to manipulate—indeed this might make for an interesting future project. Our results, however, indicate that it would be unlikely for a single manipulator to persist in a population because it either fails to invade or it invades but drives the other types that it manipulates extinct.

In terms of Figure 5 we showed the results for a single set of production terms to build intuition. We could produce similar plots for other production terms, but they would look very similar. The key to the paradox’s explanation has to do with the relationship between the relative frequencies of the cell types and the population growth rate. We have tried to improve our explanation of paradox 3 in the manuscript.

We believe that with the above additions and new results, the inclusion of paradox 3 is insightful and adds to the value of the overall paper, but we are also willing to remove it if it is too much of a distraction.

5) For the other two paradoxes, it would be nice to phrase them in evolutionary terms. Are the authors saying that a higher efficiency mutant would be selected against? If yes, this would be a way to strengthen the paradoxes. If not, then we go back to some of the issues in 1).

Response: Our new meta-population model provides an evolutionary context for our paradoxes. For paradox 1, higher efficiency mutants continually invade. They raise the

population size but decrease the relative frequency of the mutant. For paradox 2, mutants with higher efficiency either don't invade (because they lower the growth rate of type 1 cell-type 2 cell mutant pairs too much) or invade (because the type 2 cell mutant-type 2 cell mutant pairings have a high enough growth rate). We believe that this provides an evolutionary interpretation for what is occurring in our paradoxes.

REVIEWERS' COMMENTS:

Reviewer #1 (Remarks to the Author):

The authors have gone to some lengths to implement changes in response to my comments, which is commendable. The impression remains that the effects seen are only applicable to a quite specific parameter space e.g. the meta-population model only considers the case where there are two cells founding each group. Nevertheless, I appreciate the approach and the novelty of the discussion and can see that this will be a thought provoking piece for the community.

Reviewer #2 (Remarks to the Author):

I believe that the authors have addressed the concerns / questions posed by me and the other reviewers. In particular, I appreciate the demonstration that many qualitative conclusions are robust to different assumptions within the model.

One last question for the authors: In Fig 1a there is a thick line that show coexistence. Does this mean that there is a finite range of parameters for which there is true coexistence? (ie a stable fixed point that the system will go to from a range of starting abundances) I would find this surprising, but I haven't thought about it very deeply. If this is just a transition from one winning to another winning then the authors shouldn't call it coexistence and should have a thin line. I trust the authors to think about this and do what is correct / appropriate.

Reviewer #3 (Remarks to the Author):

The demonstration that adjustment of production rates through evolution leads to the same equilibrium and the addition of the meta-population model with evolution greatly strengthened the paper. With that the authors have addressed the main points raised in my initial revision and I am pleased to recommend this paper for publication.

Just a minor comment: In relation to Figs 6d and 6e: it is not only the invasion of mutant ++ (which decreases population size) but also the lack of invasion of mutant+ that illustrates paradox 2. Lack of invasion by a more efficient strain is surprising and stems from paradox 2. A sentence clarifying that can be added to the corresponding paragraph in the results.

We thank the referees and editor again for reviewing our manuscript. Here, we address the final comments of the reviewers (reviewer comments in blue font).

REVIEWER 1 COMMENTS

The authors have gone to some lengths to implement changes in response to my comments, which is commendable. The impression remains that the effects seen are only applicable to a quite specific parameter space e.g. the meta-population model only considers the case where there are two cells founding each group. Nevertheless, I appreciate the approach and the novelty of the discussion and can see that this will be a thought provoking piece for the community.

Response: Thank you for reviewing our manuscript. We agree that we only consider a small sample of possible population structures, i.e. patches are colonized by two starting cells. A more exhaustive consideration would certainly have been interesting but would have greatly exceeded the purpose of the evolutionary models, which was to show the paradoxical behavior within a more dynamic environment/population structure.

REVIEWER 2 COMMENTS

I believe that the authors have addressed the concerns / questions posed by me and the other reviewers. In particular, I appreciate the demonstration that many qualitative conclusions are robust to different assumptions within the model.

One last question for the authors: In Fig 1a there is a thick line that show coexistence. Does this mean that there is a finite range of parameters for which there is true coexistence? (i.e. a stable fixed point that the system will go to from a range of starting abundances) I would find this surprising, but I haven't thought about it very deeply. If this is just a transition from one winning to another winning then the authors shouldn't call it coexistence and should have a thin line. I trust the authors to think about this and do what is correct / appropriate.

Response: Thank you for reviewing our manuscript. We understand your concern about the line. It is supposed to be a thin line indicating a transition from one cell type winning to the other type winning. We would like to continue calling it coexistence because on that line the two strategies coexist—this is similar to coexistence lines on phase diagrams which are precisely the transition from one phase winning to the other phase winning. We have thinned the line to make this point clearer and added: “The coexistence line represents a transition between the different cell types winning.” to the figure caption for clarity.

REVIEWER 3 COMMENTS

The demonstration that adjustment of production rates through evolution leads to the same equilibrium and the addition of the meta-population model with evolution greatly strengthened the paper. With that the authors have addressed the main points raised in my initial revision and I am pleased to recommend this paper for publication.

Just a minor comment: In relation to Figs 6d and 6e: it is not only the invasion of mutant ++ (which decreases population size) but also the lack of invasion of mutant+ that illustrates paradox 2. Lack of invasion by a more efficient strain is surprising and stems from paradox 2. A sentence clarifying that can be added to the corresponding paragraph in the results.

Response: Thank you for reviewing our manuscript. You are absolutely right about the fact that the lack of invasion by a more efficient strain stems from paradox 2. We now include the sentence “This is indicative of paradox 2 because even though the + mutant is more efficient, it has a lower growth rate when combined with type 1 cells than its less efficient ancestor.” in our description of Figs 6d and 6e.